

# Phylogeny and divergence time estimation of Io moths and relatives (Lepidoptera: Saturniidae: *Automeris*)

Chelsea Skojec[1,2], Chandra Earl[3], Christian D. Couch[1,2], Paul Masonick[1] and Akito Y. Kawahara[1,2]

[1] McGuire Center for Lepidoptera and Biodiversity, Florida Museum of Natural History, University of Florida, Gainesville, FL, United States of America
[2] Department of Biology, University of Florida, Gainesville, FL, United States of America
[3] Bishop Museum, Bernice Pauahi, Honolulu, HI, United States of America

## ABSTRACT

The saturniid moth genus *Automeris* includes 145 described species. Their geographic distribution ranges from the eastern half of North America to as far south as Peru. *Automeris* moths are cryptically colored, with forewings that resemble dead leaves, and conspicuously colored, elaborate eyespots hidden on their hindwings. Despite their charismatic nature, the evolutionary history and relationships within *Automeris* and between closely related genera, remain poorly understood. In this study, we present the most comprehensive phylogeny of *Automeris* to date, including 80 of the 145 described species. We also incorporate two morphologically similar hemileucine genera, *Pseudautomeris* and *Leucanella*, as well as a morphologically distinct genus, *Molippa*. We obtained DNA data from both dry-pinned and ethanol-stored museum specimens and conducted Anchored Hybrid Enrichment (AHE) sequencing to assemble a high-quality dataset for phylogenetic analysis. The resulting phylogeny supports *Automeris* as a paraphyletic genus, with *Leucanella* and *Pseudautomeris* nested within, with the most recent common ancestor dating back to 21 mya. This study lays the foundation for future research on various aspects of *Automeris* biology, including geographical distribution patterns, potential drivers of speciation, and ecological adaptations such as antipredator defense mechanisms.

## INTRODUCTION

Saturniidae, commonly known as wild silkmoths, are renowned for their remarkable morphological features, such as large body size, vibrant wing patterns, and distinctive hindwing markings. According to *Kitching et al. (2018)*, this family comprises a staggering 3,454 species, divided into eight subfamilies and 180 genera. The subfamily Hemileucinae includes the buck and io moths and contains 51 genera. Within Hemileucinae, *Leucanella* and *Pseudautomeris* are known for their elaborate eyespots and cryptic forewings. These complex colors and patterns are distinct from other related hemileucines, like the genus *Mollipa*, with species containing less complex eyespots or lacking them entirely. The genus *Automeris* (subfamily Hemileucinae) is the most diverse genus within the family. These

Corresponding author
Chelsea Skojec, Chelseaskoj@ufl.edu

species can be found across a vast geographic range, from North America to Peru. They thrive in diverse biotopes, spanning from tropical rainforests to arid habitats, in a wide range of altitudinal levels, from the sea level up to 4000 m (*Decaens & Herbin, 2002*).

One distinctive feature of almost all *Automeris* species, is their recognizable wing ornamentation on both the fore- and hindwings. Forewings typically exhibit drab coloration and a cryptic leaf-like pattern. In stark contrast, hindwings have vibrant colors and feature an eyespot. Eyespots on the hindwings of *Automeris* moths make up the conspicuous coloration component of a deimatic display—thought to deter possible predators by startling, frightening, or confusing them, causing predators to pause or abandon their pursuit (*Umbers & Mappes, 2016*; *Drinkwater et al., 2022*). The presence of eyespots in most species of *Automeris* suggests they confer a selective benefit against predation (*Olofsson et al., 2013*), although eyespots of a few species are minimized or vestigial, which may be a secondary evolutionary loss. In Papillionoidea (butterflies), there has been extensive research on molecular mechanisms of eyespot development (*Monteiro et al., 2013*; *Nijhout, 2017*; *Matsuoka & Monteiro, 2021*) and some studies have examined the evolutionary origins and genes involved in eyespots in a few representative species of Lepidoptera (*Kodandaramaiah, 2011*; *Monteiro, 2015*; *Beldade & Monteiro, 2021*; *Sourakov & Shirai, 2020*; *Skojec, Godfrey & Kawahara, 2024*) but research into the evolution of eyespots in *Automeris* and relatives has yet to be conducted. The evolution of eyespot size and shape is believed to be driven by adaptive pressures instead of developmental constraints, thus they are not likely to constrain adaptive radiation of size, shapes, and patterns of eyespots (*Beldade, Koops & Brakefield, 2002*; *Evans & Marcus, 2006*).

Historically, researchers have relied on wing pattern variation (*e.g.*, venation, shape, size, color and pattern) to establish systematic hypotheses and determine relationships within *Automeris* (*Tuskes & McElfresh, 1995*; *Tuskes, Tuttle & Collins, 1996*). In his extensive study of the subfamily Hemileucinae, *Lemaire (2002)* identified 145 species within Automeris and classified nine species groups based on their physical appearance and genital morphology. Despite being a charismatic and extremely diverse genus, knowledge regarding the evolutionary relationships among *Automeris* species and the timing of the group's divergence from other hemileucine lineages remains limited to hypotheses from morphological characteristics. While some studies have included *Automeris* and other hemileucine species in large-scale phylogenetic analyses of Lepidoptera (*Kitching et al., 2018*; *Kawahara et al., 2019*), a comprehensive phylogenetic analysis focused on *Automeris* is yet to be conducted.

In this study we present the most comprehensive *Automeris* phylogeny that includes 80 of the 145 described species and six subspecies (*Lemaire, 2002*). Additionally, we include two morphologically similar hemileucine genera—*Pseudautomeris* (five) and *Leucanella* (four), and a morphologically distinct hemileucine genus, *Molippa* (six), to investigate relationships of *Automeris* with these genera. By constructing a well-represented phylogeny, we aim to uncover the diversification patterns within the genus and shed light on the timing of its evolutionary divergence from other lineages. Furthermore, a detailed phylogeny will provide a framework for future studies on various aspects of *Automeris* biology, including their anti-predatory defense, ecological adaptations, geographical distribution patterns,

and potential drivers of speciation. Portions of this text were previously published as part of a preprint (*Skojec, Godfrey & Kawahara, 2024*).

## MATERIALS AND METHODS

### Taxon sampling and extraction

We sampled and extracted 115 species and subspecies of *Automeris* (98), *Leucanella* (five), *Pseudautomeris* (six), and *Molippa* (six) available in the McGuire Center for Lepidoptera and Biodiversity (MGCL), Florida Museum of Natural History (FLMNH). Specimens were obtained from two collection types: dry, pinned specimens and those stored in ethanol at −80 °C, which have been wing-vouchered following *Cho et al. (2016)* and are specifically stored for use in molecular studies.

One specimen per taxon was selected from the collections. Pinned specimens were carefully selected based on their condition, focusing on those with intact wings and body parts. Priority was given to more recently collected specimens during the selection process to enhance the likelihood of successful DNA extraction. Two legs were pulled from each sample and placed in a 1.5 mL microcentrifuge tube. Corresponding identification labels for samples were recorded on the respective tubes, linking them to those collection data present in the MGCL specimen database. Similarly, molecular cold storage samples were taken from their respective boxes, and partial thoracic tissue was dissected and placed in a labeled 1.5 mL microcentrifuge tube, and all relevant information was recorded in the collective database. The extracted thorax tissue was then placed in a labeled 1.5 mL microcentrifuge tube, and all relevant information was recorded in a collective database. DNA extraction of legs from the MGCL pinned collection or thorax tissue from the molecular collection was chosen based on availability of species within the MGCL pinned and molecular collection. Due to the likelihood of higher DNA yield in thorax tissue, specimens from the molecular collection were given priority. If specimens were available in both molecular and pinned collections, specimens were chosen from the molecular collection.

Extractions were completed following the specified protocol of the Qiagen DNeasy Blood and Tissue kit (Qiagen, Hilden, Germany). For quality control, extracted DNA was checked for concentration and fragmentation with a Qubit 2.0 fluorometer and electrophoresis gels (Fisherbrand Electrophoresis Power Supplies, FB200; Thermo Fisher Scientific, Waltham, MA, USA). After DNA extraction, if the desired concentration of DNA was not achieved, samples were re-extracted if tissue was still available. Otherwise, species with multiple samples were re-extracted with a different individual. Once the quantity of DNA was determined (8 ng/uL minimum and 100 ng/uL maximum), extracted samples were sent for Anchored Hybrid Enrichment (AHE) library preparation at RAPiD Genomics in Gainesville, Florida, USA. Higher concentration samples were prioritized. Out of the 115 extracted samples, 113 showed DNA concentration of ≥8 ng/uL. Library preparation, hybridization enrichment, and Illumina HiSeq 2500 sequencing (PE100) was carried out at RAPiD Genomics.

## DNA sequencing and assembly

We used the "BOM1" Anchored Hybrid Enrichment probe set (*Hamilton et al., 2019*), which was developed to target 921 loci across Bombycoidea. We used this probe set because it has been proven effective in capturing sequence data from both dry-pinned and ethanol-preserved specimens (*Hamilton et al., 2019*; *Dowdy et al., 2020*; *Li et al., 2022*) This probe kit includes 58 loci across 24 "legacy" Sanger sequencing-based genes (*Regier et al., 2008*), eight bombycoid vision-related loci, and 855 loci designated as the Lepidoptera Agilent Custom SureSelect Target Enrichment "LEP1" probe kit from *Breinholt et al. (2018)*. Anchored Hybrid Enrichment (AHE) is a sequencing technique specifically developed to target and capture a large number of orthologous loci from the genome. This method is well-suited for resolving evolutionary relationships, both at deep and shallow levels. The probes used in this technique bind to conserved regions flanked by variable regions that are distributed randomly throughout the genome (*Hamilton et al., 2019*). This method generates a varied and informative set of loci, containing exons, introns, intergenic, and conserved regions of the genome (*Lemmon, Emme & Lemmon, 2012*).

Raw AHE sequences were assembled following the methods of *Breinholt et al. (2018)*, which implements an Iterative Baited Assembly (IBA) approach. This involves using the original sequencing probes to identify matching raw reads, which are then assembled into novel probes. The newly assembled probes are subsequently used as a query to match against the remaining raw reads, and the process is repeated iteratively until confident assemblies can no longer be obtained. The pipeline also checks for quality and cross contamination due to barcode leakage and removes paralogs. Resultant assemblies extend beyond the boundaries of the initial sequencing probes, thereby leading to the production of two distinct datasets—one comprising sequences solely from the probe region of the assembly (Probe dataset), and the other comprising sequences from the complete assembly, encompassing both the probe and outer flanking regions (PF dataset).

Because the IBA approach often resulted in multiple assembled sequences for each locus per specimen, sequences were aligned using MAFFT v7.245 (*Katoh & Standley, 2013*) and a 50% consensus generated using FASconCAT-G v1.02 (*Kück & Longo, 2014*) with the '-c -c -c' command to result in one sequence per locus per specimen. To minimize the extent of missing data in the final concatenated dataset, loci that were only obtained from three or fewer species were excluded from the datasets. Loci were concatenated across all species into one supermatrix using FASconCAT-G with the '-s' command.

A total of 113 specimens were successfully sequenced with AHE. Four outgroup species were chosen to provide secondary calibrations for the divergence time analysis and to provide a root for the tree. All outgroups chosen were genera included in the *Hamilton et al. (2019)* phylogeny and all had available raw transcriptomes or genomes. We chose two outgroup taxa in Saturniidae, *Attacus atlas* and *Therinia lactucina*, one from Sphingidae, *Manduca sexta*, and one from Bombycidae, *Bombyx mori* (*The International Silkworm Genome Consortium, 2008*; *Breinholt & Kawahara, 2013*; *Kawahara & Breinholt, 2014*; *Kanost et al., 2016*). The two transcriptomes (Saturniidae) and two genomes (*Bombyx* and *Manduca*) that we used for the outgroups were assembled to the BOM1 AHE probe regions using the methods described above. This enabled us to combine the transcriptome and
genome datasets with our newly sequenced data that were also assembled to the BOM1 AHE probe regions.

Alistat v1.6 (*Wong et al., 2020*) was used to evaluate the completeness of the concatenated alignment for Probe and PF datasets. We created Probe only and PF datasets because PF datasets can sometimes yield greater robustness to phylogenetic analyses of AHE data (*e.g.*, *Kawahara et al., 2018*; *Hamilton et al., 2019*; *Homziak et al., 2019*; *St Laurent et al., 2021*). Completeness scores (C-scores) were computed for each taxon (Cr), and those taxa with a C-score <0.15 were removed to avoid specimens with poor capture quality. We also removed loci that were captured across ≤ 3 taxa to improve dataset quality.

## Phylogenetic analysis

Phylogenetic inference was conducted on concatenated supermatrices using a maximum likelihood analysis with IQTREE *v*. 1.5 (*Nguyen et al., 2015*). We constructed two datasets, one with just the probe region, and another with both probe and PF regions. Both datasets were analyzed as nucleotides, and we determined the best substitution model and partitioning scheme using the command '-m MFP+MERGE' using ModelFinder (*Lanfear et al., 2012*) as implemented in IQTREE. The command '-B 1000 -bnni' was used to perform 1000 ultrafast bootstrap (UFBS) replicates, while optimizing each bootstrap tree using a hill-climbing nearest neighbor interchange (NNI) search to reduce the risk of overestimating branch supports. All trees were rooted to *Bombyx mori*. We refer to high node support as those with UFBS ≥95.

## Divergence time estimation

We performed divergence time estimation in BEAST v2.6.7 (*Bouckaert et al., 2019*) using the topology generated by IQTREE from the PF dataset as the starting tree. This specific topology was used since it yielded higher bootstrap values than those obtained using the Probe dataset. However, when running BEAST using the PF dataset for dating, the resulting 95% confidence intervals were disproportionately small, indicating the high likelihood that the large dataset caused the underlying Bayesian analysis to become easily stuck at a local minima (*Kawahara et al., 2019*; *Rougerie et al., 2022*). Due to this, we ultimately chose to use the Probe dataset for dating.

Loci were partitioned based on the best partitioning scheme as previously identified by ModelFinder in IQTREE and their corresponding site models (herein all set to the HKY substitution model) unlinked. For this analysis we opted to link the clock model (a relaxed clock with a log normal distribution) across all partitions because analyses running the dataset with unlinked clock models failed to converge particularly with regard to the resulting estimated divergence times. We applied a mean clock rate of 0.41 substitutions per site per 100 million years with "Mean In Real Space" checked based on the mutation rate that was recently estimated for Bombyx mori by *Han et al. (2023)* and set the 'S' parameter of our clock model prior to 0.1 so that the log normal distribution would closely match their reported 95% confidence interval of $0.33 \times 10-8$–$0.49 \times 10-8$ per site per generation (we treat a generation as one year).

Among insects, there are disproportionately few Lepidoptera fossils (*Labandeira & Sepkoski, 1993*). Therefore, our tree was calibrated using ranges of dates obtained

**Table 1 Major clade confidence intervals.** Minimum 5% confidence interval (CI), median, and maximum 95% CI ages of major clades and outgroup taxa in Millions Years Ago (mya). Node numbers correspond to those in Fig. 1.

| Clades | Node number | Minimum—5% | Median | Maximum—95% |
|---|---|---|---|---|
| *Automeris* Clade A | 1 | 0.1327 | 0.1525 | 0.1717 |
| *Pseudautomeris* | 2 | 0.0970 | 0.1186 | 0.1427 |
| *Leucanella* | 2 | 0.0970 | 0.1186 | 0.1427 |
| *io* Clade B | 3 | 0.1469 | 0.1715 | 0.1916 |
| *Molippa* | 4 | 0.1830 | 0.2102 | 0.2381 |
| *Attacus atlas* | 5 | 0.3426 | 0.3994 | 0.4558 |
| *Therinia lactucina* | 6 | 0.4426 | 0.5007 | 0.5589 |
| *Manduca sexta* | 7 | 0.5584 | 0.6188 | 0.6803 |
| *Bombyx mori* | 8 | 0.6026 | 0.6702 | 0.7370 |

from *Kawahara et al. (2019)*. We used four secondary calibration points with normal distributions to constrain the most recent common ancestors (MRCAs) of Bombycidae + (Sphingidae + Saturniidae) (the root node), Sphingidae + Saturniidae (node 7), Oxyteninae + remaining Saturniidae (node 6), and the MRCA of Saturniinae + Hemileucinae (node 5) (see Table 1). We used the Calibrated Yule model of speciation (*Yule, 1924*) as the tree prior, and ran two independent MCMC chains for 40 million generations each, sampling every 1,000 generations. TRACER v1.6.0 (*Rambaut & Drummond, 2013s*; *Rambaut & Drummond, 2013b*) was used to assess stationarity of and convergence between runs. Trees were then combined across runs with LogCombiner v2.6.3 after discarding a conservative 50% burn-in and a maximum clade credibility tree with median heights was recovered with TreeAnnotator v2.6.3 (*Rambaut & Drummond, 2013s*; *Rambaut & Drummond, 2013b*) from the posterior sampling of trees. All pipeline steps and phylogenomic analyses were conducted on the University of Florida HiPerGator high-performance computing cluster (http://www.hpc.ufl.edu/).

## RESULTS AND DISCUSSION

We were able to sample 98 species and subspecies of *Automeris*, six species of *Molippa*, five species of *Leucanella*, and six species of *Pseudautomeris*, resulting in a total of 115 sampled specimens. Of the 115, 92 were dry-pinned specimens and 23 were ethanol-stored specimens. The resulting DNA concentration range was 0.158-133 ng/uL. Samples that passed the DNA concentration cutoff (>8 ng/µL) were included in further analyses and sequencing.

In total, 113 newly sequenced samples had sufficient data for sequence assembly. We also supplemented these sequences with 4 outgroup samples, resulting in a total of 117 taxa that were included in the beginning data matrix. After removing low quality sequences from each dataset, the Probe dataset contained 589 loci across 150,646 nucleotide sites, covering 105 taxa and the PF dataset contained 906 loci across 106 taxa. The average length of each locus, as well as the percentage of missing data, were observed to be 256 nucleotides and
25% for the Probe dataset, and 2,236 nucleotides and 84% for the PF dataset, respectively. The high missing data percentage of the PF dataset is due to the nature of the flanking regions. These regions differ vastly among taxa, since their length depends on the coverage and quality of raw reads across unconserved areas.

Both probe and PF topologies support the monophyly and general placement of genera *Molippa*, *Leucanella* and *Pseudautomeris* and the paraphyly of *Automeris*. All taxa represented by multiple subspecies were recovered as monophyletic with strong support. Considering the methodology employed and the higher support values, we favor the phylogeny derived from the PF dataset (Fig. 1). However, it is important to note that both phylogenies exhibit nearly identical topologies, differing only in a few relationships at the tips while maintaining the same backbone relationships.

Recently, *Rougerie et al. (2022)* reconstructed a phylogeny of Saturniidae, which was the only modern analysis providing a hypothesis of relationships of *Automeris*, *Leucanella*, *Pseudautomeris* and *Molippa*. We find general congruence with the relationships uncovered by *Rougerie et al. (2022)*, although our study contained more species of *Automeris*. There was strong support in our study for a monophyletic *Leucanella* + *Pseudautomeris* group, which splits *Automeris* into two distinct groups—a larger clade (*Automeris* Clade A) branching around 15 million years ago (mya) and a smaller clade, including *Automeris io* (*io* Clade B), branching earlier at approximately 17 mya (see Table 2). Notably, none of the species within the smaller clade were included in the analysis by *Rougerie et al. (2022)*. However, the relationships between the species included in both studies were found to be consistent. We also found *Molippa* to be monophyletic, emerging as the sister group (UFBS = 100) to the remaining Hemileucines. This placement suggests that the most recent common ancestor of this clade (*Automeris* + *Leucanella* + *Molippa* + *Pseudautomeris*) dates to approximately 21 mya. This timeframe aligns closely with the findings of *Rougerie et al. (2022)*, who estimated a divergence of approximately 22 mya. It should be noted that our analyses specifically targeted four genera within a subfamily characterized by significant diversity, species complexes and apparent paraphyly. However, to gain a comprehensive understanding of relationships of these genera within and among the rest of the subfamily, broader sampling would be beneficial.

Eyespots of many *Leucanella* and *Pseudautomeris* are elaborate in their shape and color, and we hypothesized that these two genera are more closely related to *Automeris* than *Molippa*, as the latter have smaller, drab, potentially vestigial eyespots, or lack eyespots entirely. Our tree revealed that both *Leucanella* and *Pseudautomeris* are nested within *Automeris* with strong branch support. We postulate that predation pressure in the Neotropics drove the diversification and complexity of eyespots in species in this clade. Like many effective anti-predatory traits, eyespots in this group are likely under positive selection due to a selective benefit for survival. Diversification and complexity of eyespots may have been driven by predation pressure, given the observed anti-predatory deimatic display of eyespots in many *Automeris* species. Previous research suggests that eyespots may be adaptive in some geographic regions, and maladaptive in others, which may explain secondary losses of eyespots in *Molippa* sp. (*Kodandaramaiah, 2011*). There may be greater

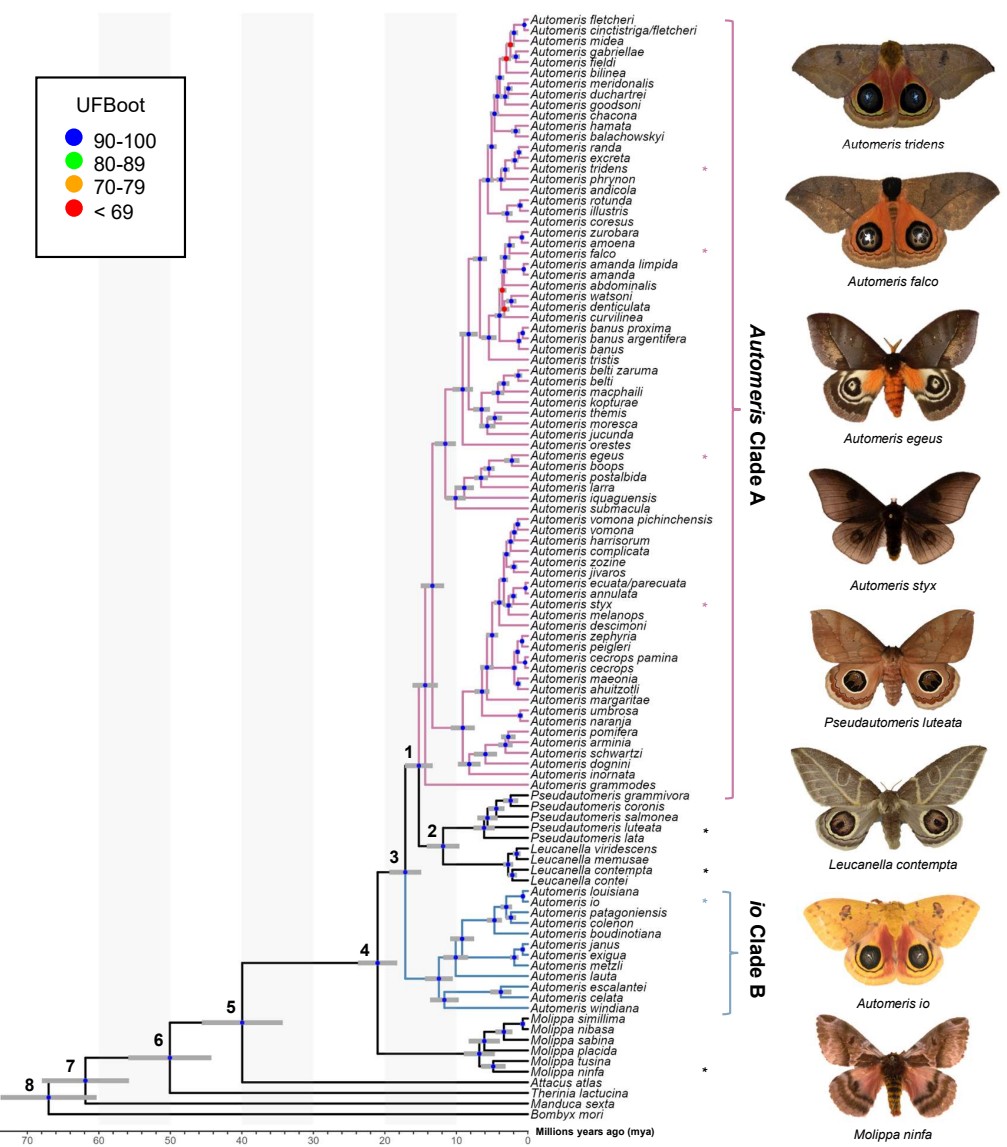

**Figure 1 Phylogeny of Automeris and relatives.** Time calibrated tree showing relationships of Automeris species and relatives, *Leucanella*, *Pseudautomeris, Molippa* and outgroups. Blue circles represent nodes where UFBoot ≥90, Red circles represent nodes where UFBoot ≤69. Divergence dates with 95% confidence intervals for each labeled node are in Table 1. Asterisks (*) notate pictured species. Photo credit: Lawrence Reeves & the McGuire Center for Lepidoptera and Biodiversity.

selective pressure on eyespots in the Neotropics than eyespots in North American species, driving the elaborate features and colors in these species (*Janzen, 1985*).

## CONCLUSION

Using Anchored Hybrid Enrichment techniques and analyses, we generated a robust phylogeny encompassing 106 taxa across *Automeris* and three closely related genera— *Leucanella*, *Pseudautomeris* and *Molippa*. This analysis reveals that *Leucanella* and

**Table 2  Normal distribution dating parameters.** These values are derived from the most recent common ancestor node and between each of the species listed and Hemileucinae subfamily from *Kawahara et al. (2019)* and are listed in terms of 100 million years. Node numbers correspond to those in Fig. 1.

| Node | Species | 5% quantile | Mean | Standard deviation | 95% quantile |
|------|---------|-------------|------|---------------------|--------------|
| 5 | *Attacus atlas* | 0.3520 | 0.4253 | 0.0444 | 0.4980 |
| 6 | *Therinia lactucina* | 0.4320 | 0.5114 | 0.0480 | 0.5900 |
| 7 | *Manduca sexta* | 0.5840 | 0.6614 | 0.0473 | 0.7390 |
| 8 | *Bombyx mori* | 0.6260 | 0.7053 | 0.0483 | 0.7850 |

*Pseudautomeris* are nested within *Automeris* with robust branch support, supporting the paraphyly of *Automeris* and suggesting close evolutionary relationships between these genera. Though this study helps clarify part of the complex Hemileucinae subfamily, a more complete sampling across species would provide greater understanding of the evolutionary patterns and processes that led to the larger diversification and evolutionary drivers of the subfamily. We hope this phylogeny will serve as a foundational framework for future investigations into the evolutionary dynamics and ecological adaptations of *Automeris* and its closely related genera. Future studies should focus on investigating eyespot trait morphology, to further clarify the diversification across species within the genus and sister groups.

## ACKNOWLEDGEMENTS

We thank the McGuire Center for Lepidoptera and Biodiversity, and the collection managers for access to specimens in the collection. We thank Nick Homziak for his comments and suggestions on the manuscript, Lawrence Reeves who took some of the images in Fig. 1, and Amanda Markee for help with specimen extraction.

### Funding

This material is based upon work supported by the National Science Foundation Graduate Research Fellowship under Grant No. 00130513 to Chelsea Skojec. Sequencing for this project was supported by the National Science Foundation DEB 1557007 to Akito Y. Kawahara. The funders had no role in study design, data collection and analysis, decision to publish, or preparation of the manuscript.

### Grant Disclosures

The following grant information was disclosed by the authors:
National Science Foundation Graduate Research Fellowship: 00130513.
National Science Foundation: DEB 1557007.

### Competing Interests

The authors declare there are no competing interests.

## Author Contributions

- Chelsea Skojec conceived and designed the experiments, performed the experiments, analyzed the data, prepared figures and/or tables, authored or reviewed drafts of the article, and approved the final draft.
- Chandra Earl conceived and designed the experiments, performed the experiments, analyzed the data, prepared figures and/or tables, authored or reviewed drafts of the article, and approved the final draft.
- Christian D. Couch performed the experiments, authored or reviewed drafts of the article, and approved the final draft.
- Paul Masonick analyzed the data, authored or reviewed drafts of the article, and approved the final draft.
- Akito Y. Kawahara conceived and designed the experiments, authored or reviewed drafts of the article, and approved the final draft.

## DNA Deposition

The following information was supplied regarding the deposition of DNA sequences:

The raw sequence reads are available at NCBI: PRJNA1030045.

## Data Availability

All assembled DNA sequences, alignments, phylogenies, and specimen metadata are available at Dryad: Skojec, Chelsea et al. (2024). Phylogeny and divergence time estimation of Io moths and relatives (Lepidoptera: Saturniidae: Automeris) [Dataset]. Dryad. https://doi.org/10.5061/dryad.547d7wmf6.

The final phylogeny is available on Open Tree of Life: https://tree.opentreeoflife.org/curator/study/view/ot_2240.

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
