# Peer review of "Phylogeny and divergence time estimation of Io moths and relatives (Lepidoptera: Saturniidae: Automeris)"

_PeerJ, doi:10.7717/peerj.17365_

## Round 0.1 · original submission · Major Revisions

Dear Dr. Skojec,

After this first review round, both reviewers provided suggestions that qualify this MS as a major review. Please look closely at all suggestions made by both reviewers to improve your manuscript. Please take special care with all the suggestions and criticisms made by reviewer #1. Still, do not forget to answer the suggestions made by reviewer #2. Please do not forget to prepare a rebuttal letter informing the reviewers about all the suggestions that were implemented and explaining those that were not included in the new version of the manuscript.

Sincerely,
Daniel SIlva

·

Basic reporting

no comment

Experimental design

As a reviewer, I always examine the figures first. What struck me with figure 1 was that the 95% HPD date ranges are by far the smallest I've ever seen in any published molecular phylogenetics dating study. Upon examining the M&M section, I have some serious concerns regarding the molecular dating methodology:

line 194: "using a fixed ultrametric tree generated by IQTREE". This may be a clerical error as I don't think that IQ-tree can produce ultrametric trees, but using a fixed ultrametric tree as input for BEAST is inappropriate and would limit the program greatly in estimating its timing evolution model, causing inflated confidence of the dating ranges as the results show. Using a fixed topology is fine, but a fixed ultrametric tree is certainly not.

Line 206 and onwards: It's fine to use an external calibration, but the authors write "our tree was calibrated using fixed dates", but contradict themselves in the next line by saying "with uniform distributions". The authors need to specify exactly which nodes were calibrated, using which ranges, and the probability prior. The results from Kawahara et al. 2019 that were used as input were HPD ranges (normal distribution) so why use a uniform distribution here? I recommend presenting the information used for calibration in a table (e.g., node, age range, probability prior).

I also disagree that a strict clock model is appropriate for a group that is ~20 million years old. I realize that without calibrations in the ingroup the effect of a relaxed clock model is limited, but it would still be the appropriate model considering the age of the group.

Considering these major issues, I have not continued to review the manuscript and recommend a major revision. I will gladly review a future version of this manuscript.

Validity of the findings

see 2.

Additional comments

no comment

·

Basic reporting

Mostly clear, see "Additional comments"

Experimental design

Sound, see "Additional comments"

Validity of the findings

Valid, see "Additional comments"

Additional comments

This manuscript presents the most comprehensive phylogeny of Automeris—a highly interesting moth genus due to evolution of deimatic displays on the hindwing of many species. The phylogeny covered 60% of all described species in the genus, as well as several closely related genera, and demonstrated the paraphyly of Automeris. This work lays out a foundation for future trait-based comparative and biogeographical analysis to look at the evolution of deimatic displays.

The manuscript is mostly clearly written and the employed sequencing and bioinformatics techniques are sound. I have a few comments to help improve the clarity of the manuscript. I would also suggest the authors re-read the manuscript (especially the Results and Discussion section, which tend to contain grammatically and descriptive inconsistencies that I might not be able to pick up).

I would recommend the publication of this manuscript if my comments were satisfactorily addressed and the authors make another attempt at improving the clarity of the manuscript.

Thank you for your work:)

ZW
* * *
start of detailed comments:

L45. I found the concept of "supporting a paraphyly" hard to comprehend. Throughout, please change to "showed/ demonstrated the paraphyly of...."

L44-46: two clauses starting with "with", can break down into two sentences.

L51: Single/plural inconsistency.

L59: Somewhere in this introduction you might want to give a more generation introduction of Hemileucinae. How many genera etc...especially for the other hemileucine genera you sequenced. How many species are there in these genera? For what reason are they NOT considered Automeris by previous researchers? Do they also have deimatic displays? This might help highlight the significance of your finding that they are actually nested within Automeris.

L71: "Lepidoptera" -- you meant non butterfly Leps right? since the first half of your sentence talks about butterflies.

L78: " assumption that species most similar in appearance are most closely related"... I haven't read " Tuskes and McElfresh, 1995; Tuskes et al. 1996"—but this is certainly an over-simplification of the work of professional taxonomists. Suggest a more nuanced rephrasing.

L82: it is confusing whether this contains two or three clauses. Also reconsider whether you meant to use "between" or "among".

L89: " Pseudautomeris and Leucanella " -- I know you have this in M&M, but good to list sample number here too.

L98: " all previously unsequenced" -- unnecessary

L110: unclear what you meant by " molecular collection samples" throughout

L114: what do you mean by "determination"?

L128: what do you mean by "promising"?

L147: Here and after, you probably meant "assembled into a novel probes" (plural)

L164: instead of "in our lab", mention citable resources

L197-203: This sentence needs to be broken down into at least three sentences. I could not follow your argument.

L228-230: This sentence is unclear. (e.g. what does "they" refer to?)

L241: broken sentence

L243: " Leucanella + Pseudautomeris" group

L247: unclear sentence. What do you mean by "both" (appeared twice)

L259: typo

L271: delete "developed"

Fig 1: One suggestion to improve the clarity of this figure. Can you put colored boxes (on the tree) to indicate CladeA and B as well as the two other genera? You can just draw them in illustrator (select increased transparency) and overlay on the tree. Then, using the same color scheme, put the same colored boxes on the illustrative images.

Also, you made a nice point showing the eyespots of the group (and the lack of which in Molippa). But no photo of Pseudautomeris?

---

## Round 0.2 · accepted · Accept

Dear Dr. Skojec,

I am pleased to inform you that your manuscript has been formally accepted for publication in PeerJ. Congratulations!

Sincerely,
Daniel Silva

·

Basic reporting

The authors have accurately addressed the comments raised by reviewers and I recommend this manuscript for publication in PeerJ

Experimental design

n/a

Validity of the findings

n/a

Additional comments

n/a